



Large carbon cycle sensitivities to climate across a permafrost thaw gradient in subarctic
Sweden
Kuang-Yu Chang*,
Climate and Ecosystem Sciences Division, Lawrence Berkeley National Laboratory,
Berkeley, California, USA
William J. Riley,
Climate and Ecosystem Sciences Division, Lawrence Berkeley National Laboratory,
Berkeley, California, USA
Patrick Crill,
Department of Geological Sciences, Stockholm University, Stockholm, Sweden
Robert F. Grant,
Department of Renewable Resources, University of Alberta, Edmonton, Alberta, Canada
Virginia Rich,
Department of Microbiology, The Ohio State University, Columbus, Ohio, USA
and,
Scott Saleska,
Department of Ecology and Evolutionary Biology, University of Arizona, Tucson,
Arizona, USA
*Corresponding author: Kuang-Yu Chang, ckychang@lbl.gov
Climate and Ecosystem Sciences Division, Lawrence Berkeley National Laboratory
Berkeley, California, USA



Phone: (510) 495-8141



26                                        Abstract

Permafrost peatlands store large amounts of carbon potentially vulnerable to
decomposition. However, the fate of that carbon in a changing climate remains uncertain
in models due to complex interactions among hydrological, biogeochemical, microbial,
and plant processes. In this study, we estimated effects of climate forcing biases present
in global climate reanalysis products on carbon cycle predictions at a thawing permafrost
peatland in subarctic Sweden. The analysis was conducted with a comprehensive
biogeochemical model (*ecosys*) across a permafrost thaw gradient encompassing intact
palsa with an ice core and a shallow active layer, partly thawed bog with a deeper active
layer and a variable water table, and fully thawed fen with a water table close to the
surface, each with distinct vegetation and microbiota. Using *in situ* observations to
correct local cold and wet biases found in the Global Soil Wetness Project Phase 3
(GSWP3) climate reanalysis forcing, we evaluated our model performance by comparing
predicted and observed carbon dioxide ($CO_2$) and methane ($CH_4$) exchanges, thaw depth,
and water table depth. The simulations driven by the bias-corrected climate suggest that
the three peatland types currently accumulate carbon from the atmosphere, although the
bog and fen sites can have annual positive radiative forcing impacts due to their higher
$CH_4$ emissions. Our simulations indicate that projected precipitation increases could
accelerate $CH_4$ emissions from the palsa area, even without further degradation of palsa
permafrost. The GSWP3 cold and wet biases for this site significantly alter simulation
results and lead to erroneous active layer depth and carbon budget estimates. Biases in
simulated $CO_2$ and $CH_4$ exchanges from biased climate forcing are as large as those
among the thaw stages themselves at a landscape scale across the examined permafrost



thaw gradient. Future studies should thus not only focus on changes in carbon budget
associated with morphological changes in thawing permafrost, but also recognize the
effects of climate forcing uncertainty on carbon cycling.





## 1. Introduction

Confidence in future climate projections depends on the accuracy of terrestrial carbon budget estimates, which are presently very uncertain (Friedlingstein et al., 2014; Arneth et al., 2017). In addition to the complexity in physical process representations, a major source of this uncertainty comes from challenges in quantifying climate responses induced by biogeochemical feedbacks. Increases in atmospheric carbon dioxide ($CO_2$) concentrations can directly stimulate carbon sequestration from plant photosynthesis (Cox et al., 2000; Friedlingstein et al., 2006) and indirectly stimulate carbon emissions (e.g., from soil warming and resulting increased respiration), although the predicted magnitudes of these exchanges strongly depend on model process representations (Zaehle et al., 2010; Grant, 2013, 2014; Ghimire et al., 2016; Chang et al, 2018).

The undecomposed carbon stored in permafrost is of critical importance for biogeochemical feedbacks to climate because it is about twice as much as currently is in the atmosphere (Hugelius et al., 2014) and is vulnerable to release to the atmosphere as permafrost thaws (Schuur et al., 2015). Lundin et al. (2016) reported that it is plausible (71% probability) for the high latitude terrestrial landscape to serve as a net carbon source to the atmosphere, although its peatland components would remain atmospheric carbon sinks.

In addition to the overall carbon balance of the changing Arctic, the type of carbon gaseous emission is important to climate feedbacks. High latitudes are predicted to get wetter (IPCC, 2014), and saturated anaerobic conditions facilitate methane ($CH_4$) production, which is a much more efficient greenhouse gas than $CO_2$ in terms of global warming potential. Even habitats that can be net carbon sinks can produce positive




radiative forcing impacts on climate due to $CH_4$ release, as Bäckstrand et al. (2010)
showed for a subarctic peatland. Under projected warming and wetting trends in the
Arctic (Collins et al., 2013; Bintanja and Andry, 2017), carbon cycle feedbacks over the
permafrost region could become stronger as increased precipitation enhances surface
permafrost thaw and strengthens $CH_4$ emissions by expansion of anaerobic volume
(Christensen et al., 2004; Wickland et al., 2006).

The Stordalen Mire in northern Sweden (68.20°N, 19.05°E) is in the

discontinuous permafrost zone, encompassing a mosaic of thaw stages with associated
distinct hydrology and vegetation (Christensen et al. 2004; Malmer et al., 2005),
microbiota (Mondav and Woodcroft et al., 2014; Mondav et al., 2017; Woodcroft and
Singleton et al., 2018), and organic matter chemistry (Hodgkins et al., 2014). These
landscapes have been shifting over the last half-century to a more thawed state, likely due
to recent warming (Christensen et al. 2004). Drier hummock sites dominated by shrubs
have degraded to wetter sites dominated by graminoids (Malmer et al., 2005; Johansson
et al., 2006). The thaw-induced habitat shifts are associated with increases in landscape
scale $CH_4$ emissions (Christensen et al. 2004; Johansson et al., 2006; Cooper et al., 2017)
reflective of the higher $CH_4$ emissions of the wetter thawed habitats (McCalley et al.,
2014). The higher $CO_2$ uptake in later thaw-stage habitats has not compensated for the
increase in positive radiative forcing from elevated $CH_4$ emissions (Bäckstrand et al.,
2010; Deng et al., 2014).

The impacts of climate sensitivity on the terrestrial carbon cycle have been

investigated at the global scale, and the results highlight the need to consider uncertainty
in climate datasets when evaluating permafrost region carbon cycle simulations



(Ahlström et al., 2017; Guo et al., 2017; Wu et al., 2017). Ahlström et al. (2017) showed
that climate forcing biases are responsible for a considerable fraction (~40%) of the
uncertainty range in ecosystem carbon predictions from18 Earth System Models (ESMs)
reported by Anav et al. (2013). Guo et al. (2017) concluded that the differences in climate
forcing contribute to significant differences in simulated soil temperature, permafrost
area, and active layer thickness. Wu et al. (2017) demonstrated that differences among
climate forcing datasets contributes more to predictive uncertainty than differences in
apparent model sensitivity to climate forcing. However, notably, none of these studies
accessed the effects on $CH_4$ emissions, and their spatial resolution could not represent
site-level spatial heterogeneity observed in arctic tundra (Grant et al. 2017a; 2017b).

Here, we use the ecosystem model *ecosys*, which employs a comprehensive set of

fully coupled biogeochemical and hydrological processes, to estimate the effects of
climate forcing uncertainty and sensitivity on $CO_2$ and $CH_4$ exchanges and active layer
thickness simulations. For the Stordalen Mire site, we estimated bias in the Global Soil
Wetness Project Phase 3 (GSWP3) climate reanalysis dataset using site-level long-term
meteorological measurements and evaluated impacts on simulated soil and plant
processes across the permafrost thaw gradient. This approach enables us to assess model
sensitivity to individual climate forcing biases, instead of the aggregated uncertainty
range embedded in climate datasets (e.g., variations of climate conditions represented in
different climate datasets) presented in previous studies. We address the following
questions for our study site at the Stordalen Mire: (1) What are the biases embedded in
the GSWP3 climate reanalysis dataset? (2) How do those biases affect model predictions
of active layer depth, $CO_2$ exchanges, and $CH_4$ exchanges? (3) How does climate




sensitivity vary across the stages of permafrost thaw? In addition to improving
understanding of permafrost responses to climate, we identify ecosystem carbon
prediction uncertainty induced by climate forcing uncertainty in general as the biases
found in GSWP3 were consistent with other climate reanalysis datasets during the last
decade (section 3).

**2.  Methods and Data**
**2.1 Study site description**
Our study sites are located at the Stordalen Mire (68.20 °N, 19.03 °E: 351 m
above sea level), which is about 10 km southeast of the Abisko Scientific Research
Station (ANS) in northern Sweden. Significant changes in climate over this region have
been recorded during the last few decades. The annual mean air temperature measured at
the ANS has risen by 2.5 °C from 1913 to 2006, where it exceeded the 0 °C threshold
(0.6 °C in 2006) for the first time over the past century (Callaghan et al., 2010). The
measured annual total precipitation has also increased from 306 mm y$^{-1}$ (years 1913 to
2009) to 336 mm y$^{-1}$ (years 1980 to 2009) (Olefeldt and Roulet, 2012), along with
increased variability in extreme precipitation (Callaghan et al., 2010). The measured
annual maximum snow depth has increased from 59 cm (years 1957 to 1971) to 70 cm
(years 1986 to 2000), and the snow cover period with snow depth greater than 20 cm has
decreased from 5.8 months (years 1957 to 1971) to 4.9 months (years 1986 to 2000)
(Malmer et al., 2005).
The Stordalen Mire can be broadly classified into three peatland types: intact
permafrost palsa, partly thawed bog, and fully thawed fen (Hodgkins et al., 2014),





hereafter referred to as palsa, bog, and fen. The spatial distribution of these peatland
types in 2000 can be found in Olefeldt and Roulet (2012). The palsa sites are
ombrotrophic and raised 0.5 to 2.0 m above their surroundings, with a relatively thin peat
layer (0.4 to 0.7 m, Rydén et al., 1980), thinner active layer depth (less than 0.7 m in late
summer), and no measurable water table depth (Bäckstrand et al., 2008a; 2008b; Olefeldt
and Roulet, 2012). The bog sites are ombrotrophic and are wetter than the palsa sites,
with a thicker peat layer (0.5 to ~1 m, Rydén et al., 1980), thicker active layer depth
(ALD) (greater than 0.9 m), and water table depth fluctuating from 35 cm to the ground
surface (Bäckstrand et al., 2008a; 2008b; Olefeldt and Roulet, 2012). The fen sites have
no permafrost and are minerotrophic, receiving a large amount of water from a lake to the
east of the mire, with water table depths near or above the ground surface (Bäckstrand et
al., 2008a; 2008b; Olefeldt and Roulet, 2012).

Differences in hydrology and permafrost conditions create high spatial

heterogeneity with different soil moisture, pH, and nutrient conditions that support
different plant communities (Bäckstrand et al., 2008a; 2008b). The palsa is dominated by
dwarf shrubs with some sedges, feather mosses, and lichens (Malmer et al., 2005;
Bäckstrand et al., 2008a; 2008b; Olefeldt and Roulet, 2012). The bog is dominated by
Sphagnum spp. mosses with a moderate abundance of sedges (Malmer et al., 2005;
Bäckstrand et al., 2008a; 2008b; Olefeldt and Roulet, 2012). The fen sites we studied are
dominated by sedges (Bäckstrand et al., 2008a; 2008b).

**2.2 Field measurements**





Continuous daily meteorological measurements have been recorded at the ANS
since 1913, including air temperature, precipitation, wind speed, wind direction, relative
humidity, and snow depth. Measurements of solar radiation, longwave radiation, and soil
temperature are also available at the ANS since 1982. The soil thaw depth (measured to
90 cm) and water table depth measurements were taken in the three peatland types 3 to 5
times per week from early May to mid-October during 2003 to 2007 (Bäckstrand et al.,
2008b).
CO$_2$ and CH$_4$ exchanges at the three peatland types were measured with
automated chambers during the thawed seasons from 2002 to 2007. Chamber lids were
removed in the Fall and replaced in the Spring. Three chambers were in the palsa, three
were in the bog, and two were in the fen. Each chamber covered an area of 0.14 m$^2$ with a
height of 25–45 cm depending on the vegetation and the depth of insertion. Each
chamber was closed for 5 minutes every 3 hours to measure CO$_2$ and total hydrocarbon
(THC) exchanges. CH$_4$ exchanges were manually observed approximately 3 times per
week, and these measurements were used to quantify the proportion of CH$_4$ in the
measured THC (Bäckstrand et al., 2008a). The CH$_4$ exchanges were near zero in the
palsa sites (Bäckstrand et al., 2008a; Bäckstrand et al., 2008b; Bäckstrand et al., 2010), so
it was not incorporated in our model evaluation. We used the CO$_2$ and CH$_4$ exchanges
observed at 3-hourly steps when the R$^2$ values recorded in the measurements were greater
than 0.8 (Tokida et al., 2007), and then calculated the associated daily mean exchanges
when there were 8 measurements per day (Table 1). The quality-controlled daily
measurements only covered 12.4–33.7% of the daily data points because of the lack of
continuous quality-controlled 3-hourly measurements. The data screening was applied to



exclude unreliable measurements and avoid biases from inappropriate gap filling, which
is necessary for model evaluations. More detailed descriptions of the $CO_2$ and $CH_4$
exchanges measurements can be found in Bäckstrand et al. (2008a).

**2.3 GSWP3**

GSWP3 is an ongoing modeling activity that provides global gridded

meteorological forcing (0.5° x 0.5° resolution) and investigates changes in energy-water-
carbon cycles throughout the 20th and 21st centuries. The GSWP3 dataset is based on the
20th Century Reanalysis (Compo et al., 2011), using a spectral nudging dynamical
downscaling technique described in Yoshimura and Kanamitsu (2008). A more detailed
description of the GSWP can be found in Dirmeyer (2011) and van den Hurk et al.

(2016).

In this study, we extracted the meteorological conditions at the Stordalen Mire

from 1901 to 2010 from the GSWP3 climate reanalysis dataset. The 3-hourly products of
air temperature, precipitation, solar radiation, wind speed, and specific humidity were
interpolated to hourly intervals with cubic spline interpolation to serve as the
meteorological inputs used in our model.

The GSWP3 dataset was chosen over other existing climate reanalysis datasets for

its spatial and temporal resolutions. For example, the Climatic Research Unit (CRU;
Harris et al., 2014) dataset provided monthly meteorological forcing at 0.5° x 0.5°
resolution; the National Centers for Environmental Prediction (NCEP; Kalnay et al.,
1996; Kanamitsu et al., 2002) dataset provided 6-hourly meteorological forcing at T62
Gaussian grid (~1.915° x 1.895° resolution); the CRUNCEP (Viovy, 2018) dataset



provided 6-hourly meteorological forcing at 0.5° x 0.5° resolution; and the European
Centre for Medium-Range Weather Forecasts (ECMWF; Berrisford et al., 2011) dataset
provided 3-hourly meteorological forcing with 125 km (~1.125°) horizontal resolution.

**2.4 Model description**

*Ecosys* is a comprehensive biogeochemistry model that simulates ecosystem

responses to diverse environmental conditions with explicit representations of microbial
dynamics and soil carbon, nitrogen, and phosphorus biogeochemistry. The above-ground
processes are represented in multi-layer plant interacting canopies, and the below-ground
processes are represented in multiple soil layers with multi-phase subsurface reactive
transport. *Ecosys* operates at variable time steps (down to seconds) determined by
convergence criteria, and it can be applied at patch scale (spatially homogenous one-
dimensional) and landscape scale (spatially variable two- or three-dimensional). Detailed
descriptions, including inputs, outputs, governing equations, parameters, and references
of the *ecosys* model can be found in Grant (2013).

The *ecosys* model has been extensively tested against eddy covariance fluxes and

related ecophysiological measurements with a wide range of sites and weather conditions
in boreal, temperate, and tropical forests (Grant et al., 2007a; Grant et al., 2007c; Grant et
al., 2009a; Grant et al., 2009b; Grant et al., 2009c; Grant et al., 2010), wetlands (Dimitrov
et al., 2011; Grant et al., 2012b; Dimitrov et al., 2014; Mezbahuddin et al., 2014),
grasslands (Grant and Flanagan, 2007; Grant et al., 2012a), tundra (Grant et al., 2003;
Grant et al., 2011b; Grant 2015; Grant et al., 2015), croplands (Grant et al., 2007b; Grant
et al., 2011a), and other permafrost-associated habitats (Grant and Roulet, 2002; Grant,





2017a; Grant et al., 2017b). All *ecosys* model structures are unchanged from those
described in these earlier studies.
**2.5 Experimental design**

To evaluate the effects of climate on model predictions, we conducted four sets of

simulations at each of the three peatland types at the Stordalen Mire from 1901 to 2010.
The 110 year simulations were performed to ensure the simulation was equilibrated with
local climate (Grant et al. 2017a).

The meteorological conditions for all the simulations were based on the hourly

data extracted from the GSWP3 climate reanalysis dataset (section 2.3). The monthly
mean bias of the GSWP3 for this location was calculated by comparing it to the air
temperature and precipitation measured at the ANS, for years 1913 to 2010 (section 3.1).
The full series of air temperature and precipitation extracted from GSWP3 were then
bias-corrected using the monthly mean bias calculated from 1913 to 2010; we label this
model scenario CTRL. Our bias correction was conceptually similar to the one used in
Ahlström et al. (2017), where the bias-corrected climate forcing fields were the ESM
outputs adjusted by the corresponding bias calculated from observations in a reference
period.

The simulation results from CTRL should represent the reliability of applying

*ecosys* at the Stordalen Mire because CTRL is driven by the best local climate
description. We first evaluated predicted thaw depth, water table depth, and $CO_2$ and $CH_4$
exchanges using the CTRL simulation (section 3.2 to 3.4). In the second set of
simulations, BIASED-COLD, the biased GSWP3 air temperature data was used, and we
corrected only the GSWP3 precipitation. Deviations between CTRL and BIASED-COLD



reflect biased air temperature's effects on responses across the thaw gradient. In the third
set of simulations, BIASED-WET, we bias-corrected the air temperature extracted from
GSWP3, which allows us to quantify the effects of biased precipitation. Finally, we used
the meteorological conditions directly extracted from GSWP3 to drive our fourth set of
simulations, BIASED-COLD&BIASED-WET, which reveals the uncertainty range of
subarctic peatland simulation associated with the local biases in GSWP3 climate forcing.

While the three peatland types share the same climate conditions, they differ in

soil hydrologic conditions and vegetation characteristics (section 2.1). The bulk density
and porosity profiles were set to the values reported in Rydén et al. (1980), who
suggested a decreasing trend of bulk density and an increasing trend of porosity from
palsa (0.12 Mgm$^{-3}$ at surface; 92–93% within the upper 10 cm) to bog and fen (0.06
Mgm$^{-3}$ at surface; 96–97% within the upper 10 cm). The peatland soil carbon-to-nitrogen
(C/N) ratios and pH values were assigned according to Hodgkins et al. (2014), who
documented an increasing trend of pH from palsa (4.0), to bog (4.2), to fen (5.7), and a
decreasing trend of soil organic matter C/N ratio from bog (46±18), to palsa (39±24), to
fen (19±0.4). Common values of field capacity (0.4) and wilting point (0.15) were used
for the three peatland types (Deng et al., 2014).

**3 Results and Discussion**
**3.1 GSWP3 climate comparison to observations**

As described in section 2.3, we extracted meteorological conditions at the

Stordalen Mire from the GSWP3 climate reanalysis dataset. The closest GSWP3 grid cell
was centered at 68.0 °N and19.0 °E, which covers the Stordalen Mire and the ANS. The



annual mean air temperature and precipitation calculated at this GSWP3 grid cell were -
3.65 °C and 683.88 mm y$^{-1}$, respectively, for years 1913 to 2010. A cold bias (-3.09 °C)
was identified in the GSWP3 annual mean air temperature during the 1913 to 2010
period, although a very high correlation coefficient (r = 0.99) was found when compared
with the ANS measurements (Figure 1a). Both time series exhibit an overall warming
trend from the early 20[th] century to the present (0.01°C y$^{-1}$), with an even more prominent
warming trend from 1980 to 2010 (0.05 °C y$^{-1}$ [ANS] and 0.04 °C y$^{-1}$ [GSWP3]).

Similarly, the GSWP3 annual total precipitation data correlates well with ANS

measurements (r = 0.80) but has a wet bias of 380 mm y$^{-1}$ between 1913 and 2010
(Figure 1b). An increasing trend in annual total precipitation was recorded in both time
series from the early 20[th] century to present (0.47 mm y$^{-2}$ [ANS] and 1.07 mm y$^{-2}$
[GSWP3]), although a decreasing trend was found from 1980 to 2010 (-0.56 mm y$^{-2}$
[ANS] and -2.39 mm y$^{-2}$ [GSWP3]).

The seasonal cycle of the GSWP3 monthly mean air temperature also matches

that measured at the ANS, with a very high correlation coefficient (r = 0.99; Figure 2a).
The underestimation bias and inter-annual variability of GSWP3 air temperature are
greater in winter (maximum underestimate in December, at -4.52 °C) and smaller in
summer (minimum underestimate in July, at -1.52 °C), respectively.

The magnitude and inter-annual variability of the GSWP3 monthly mean

precipitation are comparable between winter and summer, while the ANS measurements
exhibit stronger seasonality with lower magnitudes during winter. Despite the differences
found in seasonal patterns, a high correlation coefficient (r = 0.64) was found between the
monthly mean precipitation extracted from GSWP3 and the ANS measurements. The





overestimation of monthly mean precipitation was greatest in December (43.25 mm
month$^{-1}$) and smallest in August (18.75 mm month$^{-1}$).

These comparisons suggest that GSPW3 air temperature and precipitation data

reasonably capture measured seasonal and long-term trends over past decades, but are
biased cold and wet compared to observations, especially during winter. Similar cold and
wet biases exist in CRUNCEP and ECMWF climate reanalysis datasets during our 2003
to 2007 study period (Supplemental Material Figure 1). The calculated annual mean air
temperature and precipitation at the Stordalen Mire for years 2003 to 2007 were -2.49 °C
(precipitation 795.09 mm y$^{-1}$), -2.46 °C (708.60 mm y$^{-1}$), and -2.28 °C (765.67 mm y$^{-1}$) in
the GSWP3, CRUNCEP, and ECMWF climate reanalysis datasets, respectively.

**3.2 Model testing**
**3.2.1 Thaw depth**

We first evaluated *ecosys* against observations using bias-corrected climate

forcing (i.e., the CTRL simulation). Predicted thaw depth agrees well with measurements
collected from 2003 to 2007 for all examined peatland types (Figure 3), with a correlation
coefficient of 0.95, 0.87, and 0.41 at the palsa, bog, and fen, respectively. Both
simulations and observations show that the rate of thaw depth deepening in the summer
varies with peatland type (i.e., relatively slow, moderate, and rapid at the palsa, bog, and
fen sites, respectively).

Predicted and observed maximum thaw depth (i.e., Active Layer Depth, ALD) in

the intact permafrost palsa was between 45 and 60 cm in September. In the partly thawed
bog, the simulated thaw depth is slightly shallower than that observed before August. The





simulated bog thaw depth becomes greater than 90 cm by the end of August, which
matches the time when measured thaw depth reaches its maximum. The thaw depth
becomes greater than 90 cm by the end of July in the fully thawed fen. The patterns of
thawing permafrost presented here are consistent with Deng et al. (2014), who simulated
the same site using the DNDC model.

**3.2.2 $CO_2$ exchanges**
The daily Net Ecosystem Exchange (NEE) simulated in the CTRL simulation
reasonably captures observed seasonal dynamics from 2003 to 2007 for all the examined
peatland types (Figure 4). The simulations and observations showed net $CO_2$ uptake
during summer and release during winter. The observations and simulations also showed
large $CO_2$ emissions in the palsa site during Fall of 2004. Simulated Fall $CO_2$ bursts in the
three sites in other years could not be confirmed because of a lack of observations during
these periods. Similar to the patterns reported in Raz-Yaseef et al. (2016), some episodic
$CO_2$ emission pulses were simulated as surface ice thaws in Spring, but there were no
measurements to confirm those events. The correlation coefficients of the simulated and
observed daily NEE ranged from 0.58 to 0.60, and most of the discrepancies between the
simulations and observations were within the ranges of NEE variability measured at
different subsites within the same peatland type.
As described in section 2.2, simulated $CO_2$ exchanges were evaluated for 3-hourly
and daily time steps when quality-controlled measurements were available ($R^2$ values and
relative root mean squared errors (RRMSEs) shown in Table 2). Simulated NEE is in
reasonable agreement with the 3-hourly NEE measurements with RRMSEs ranging



from 8.4 to 19.1%. Model performance was generally poorer at daily time steps, although
the calculated RRMSEs were comparable to those reported in Deng et al. (2014). We
suspect this degradation resulted from uncertainty in determining a daily NEE
representative of the entire peatland type due to (1) limited daily data points (less than
14% across the study period, Table 1) due to lack of continuous quality-controlled 3-
hourly measurements and (2) the large variability of daily NEE ranges measured at
different subsites within the same peatland type (Figure 4). Our results thus indicate that
NEE is affected by thaw stage (Bäckstrand et al., 2010; Deng et al., 2014) and fine scale
spatial heterogeneity of the system. More detailed measurements with higher spatial and
temporal resolutions within the same peatland type would be necessary to characterize
the effects of this type of heterogeneity.

**3.2.3 Water table depth and CH$_4$ exchanges**

Simulated water table depth generally captures observed seasonal patterns

measured in the bog and fen sites from 2003 to 2007 (Figure 5a, c). During summer, the
predicted bog water table depth fluctuates around the ground surface, and the predicted
water table depth is at or above the ground surface in the fen. Water table depths
simulated by *ecosys* are generally higher than the corresponding measurements in the
bog, where measured water table depths are often below the ground surface with greater
seasonal variability. Simulated fen water table depths have better overall fit to
observations, being higher (~5 cm) than measurements in 2003 and 2004, close to
measurements in 2005 and 2006, and slightly deeper (~2 cm) than measurements in 2007.
The discrepancies in water table depth could be driven by the limitations of our one-

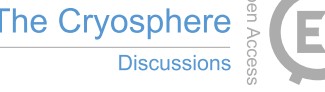



dimensional column simulation which inhibits lateral water transport and hinders the
variations of water table depth, which is a particular issue in simulating the dynamic
water table of the bog. A multi-dimensional simulation that includes realistic topographic
effects could help improve the representation of water table dynamics, and estimates of
the measurement uncertainty would help facilitate the assessment of simulation bias.

Simulated and measured daily $CH_4$ exchanges correlate reasonably well in the

bog (r = 0.49) and well in the fen (r = 0.65) sites across the study period (Figure 5b, d).
Both the simulations and observations have stronger $CH_4$ emissions during summer with
peak emissions in late summer. Some episodic $CH_4$ emission pulses (Mastepanov et al.,
2008) were simulated during shoulder seasons, and the simulated amount of post-growing
season $CH_4$ emissions agrees well with those measured in 2007.

Most of the discrepancies between simulated and observed $CH_4$ emissions were

within the variability of measurements across subsites within the same peatland type. The
3-hourly and daily RRMSEs ranged from 11.1 to 22.3% (Table 2) and the daily RRMSEs
were comparable to results presented in Deng et al. (2014). Our results show that model
evaluation of $CH_4$ emissions with finer temporal resolution observations is not necessarily
superior to evaluation with coarser temporal resolution. This result could be related to
weaker $CH_4$ emission variability measured across subsites within the same peatland type
(Figure 5b, d).

**3.3 Variability across the permafrost thaw gradient**

Thaw rate and ALD increase along the thaw gradient (i.e., palsa to bog to fen),

and landscape variations are generally greater than simulated inter-annual variability



(Figure 6a). Maximum carbon uptake also increases along the thaw gradient, and
variations across the landscape are comparable with simulated intra-seasonal and inter-
annual variabilities (Figure 6b). The simulated mean seasonal cumulative NEE were
calculated based on the seasonality identified in Bäckstrand et al. (2010), and the results
show that the magnitude of mean growing season $CO_2$ uptake is highest in the fen and
lowest in the palsa (Table 3). The same rank applies to the magnitude of mean $CO_2$
emissions over the non-growing season, although differences across the thaw gradient are
smaller.

$CH_4$ emission rates increase significantly along the thaw gradient, and the palsa

site emissions are negligible (Figure 6c). Mean cumulative $CH_4$ emissions simulated in
the fen site are much higher than those in the bog site, and most $CH_4$ emissions occur
during the growing season (Table 3). The higher $CH_4$ emissions in the fully thawed fen
can be attributed to its faster thaw rate (Figure 6a) and a water table depth close to the
surface (Figure 5c). Seasonal cumulative NEE and $CH_4$ emissions from observations
could not be accessed due to the lack of continuous quality controlled carbon flux
measurements during our study period (Table 1).

**4. Climate sensitivity of permafrost thaw**
**4.1 Thaw responses to climate**

For each of the four sets of simulations with different climate forcing (section

2.5), simulated mean ALD from 2003 to 2007 is always greatest in the fen and lowest in
the palsa (Figure 7). This consistent trend along the thaw gradient indicates that ALDs
are largely regulated by their distinct ecological and hydrological conditions, because all





three sites had the same climate forcing in each set of simulations (i.e., CTRL, BIASED-
COLD, BIASED-WET, and BIASED-COLD&BIASED-WET). Therefore, the intact
permafrost palsa, partly thawed bog, and fully thawed fen have different resilience
against the changes in climate forcing, and this type of ecosystem resilience plays an
important role in determining ALD under changes in climate conditions.

Effects of climate on simulated ALD are similar across peatland types (Figure 7).

With increased precipitation (BIASED-WET vs. CTRL), simulated ALD generally
becomes deeper with greater inter-annual variability at all the examined peatland types.
This effect is less prominent in the comparison between experiments BIASED-COLD
and BIASED-COLD&BIASED-WET, possibly because the cold biases in these two
experiments (section 3.1) constrain ALD variation. The simulated ALD also becomes
deeper with higher air temperature (CTRL vs. BIASED-COLD; BIASED-WET vs.
BIASED-COLD&BIASED-WET) at all the examined peatland types. This response is
more evident in the comparison between experiments BIASED-WET and BIASED-
COLD&BIASED-WET, probably driven by their wet bias (section 3.1) that facilities
ALD deepening (via increased thermal conductivity and advective heat transport; Grant
et al. 2017a). Similar dependencies between ALD and climate were shown in Åkerman
and Johansson (2008) and Johansson et al. (2013), based on multi-year measurements and
snow manipulation experiments.

Therefore, the combined cold and wet biases in the GSWP3 climate reanalysis

dataset could counteract their individual effects on simulated ALD development at the
Stordalen Mire. Our results indicate a 28.6%, 0.7%, and 11.7% underestimation of ALD
simulated in the palsa, bog, and fen sites, respectively, when applying the GSWP3



climate reanalysis data over this region without proper bias correction (BIASED-
COLD&BIASED-WET vs. CTRL). Our sensitivity analysis suggests that projected
warming and wetting trends (Collins et al., 2013) could significantly increase ALD in the
Arctic, since increases in precipitation and air temperature can both contribute to ALD
deepening.

**4.2 Carbon budget responses to climate**

Annual mean (from 2003 to 2007) $CO_2$ and $CH_4$ exchanges simulated with the

four climate forcing datasets (section 2.5) indicate a general $CO_2$ sink and $CH_4$ source,
except the weak $CO_2$ emissions simulated at the fen in experiment BIASED-
COLD&BIASED-WET (Figure 8a,b). Our results also indicate that differences in annual
$CO_2$ and $CH_4$ exchanges across the four climate forcing datasets for a single peatland type
are as large as those across peatland types for a single climate forcing dataset (Figure
8a,b). These large $CO_2$ and $CH_4$ exchanges climate sensitivities demonstrate that the
peatland's dynamical responses to climate have stronger effects on the carbon cycle than
on ALDs (Figure 7).

With bias-corrected precipitation, increased air temperature (CTRL vs. BIASED-

COLD) leads to stronger $CO_2$ uptake and greater $CH_4$ emissions at all the examined
peatland types (Figure 8a,b), mainly because enhanced sedge growth facilitates carbon
cycling under a warmer environment (results not shown). This air temperature sensitivity
affects $CO_2$ and $CH_4$ exchanges within the same peatland type without significantly
changing ALD (Figure 7). For both experiments, $CO_2$ uptake and $CH_4$ emissions are
greatest in the fully thawed fen and lowest in the intact permafrost palsa, consistent with



the measurements reported in Bäckstrand et al. (2010) for the same period. Based on the
Coupled Model Intercomparison Project, phase 5 (CMIP5) ESM simulations, arctic
annual mean surface air temperature is projected to increase by 8.5±2.1 °C over the 21st
century (Bintanja and Andry, 2017). This projected air temperature increase is more than
double the air temperature difference between site-observed and GSWP3 temperatures,
which could significantly enhance $CH_4$ emissions regardless of palsa degradation into bog
and fen.

On the other hand, wet biases (BIASED-WET and BIASED-COLD&BIASED-

WET) increase $CH_4$ emissions in the palsa site; wetter and colder conditions result in as
much $CH_4$ release as the current fen sites, while wetter conditions alone drive palsa
emissions comparable to the current bog sites (Figure 8b). The large precipitation
sensitivity found in palsa $CH_4$ emissions could have strong effects on palsa carbon
cycling because arctic precipitation is projected to increase by 50 – 60 % towards the end
of the twenty-first century (based on CMIP5 estimates; Bintanja and Andry, 2017). The
comparison between experiments BIASED-WET and BIASED-COLD&BIASED-WET
shows that in the palsa, increased air temperature strengthens $CO_2$ uptake and weakens
$CH_4$ emissions. This shift is primarily driven in the model by increased shrub and moss
productivity under the warmer environment, which facilitate $CO_2$ uptake while drying out
the soil and reducing $CH_4$ emissions (results not shown). In the bog and fen sites,
increased air temperature under wet bias strengthens both the simulated $CO_2$ uptake and
$CH_4$ emissions (BIASED-WET vs. BIASED-COLD&BIASED-WET), due to enhanced
sedge growth under the warmer environment that facilitates carbon cycling in the
experiment BIASED-WET.



488   We assessed the integrated effects of the changes in $CO_2$ and $CH_4$ exchanges

489 identified in the full suite of simulations in terms of the Net Carbon Balance (NCB) and

490 net emissions of greenhouse gases expressed as $CO_2$ equivalents (Net Greenhouse Gas

491 Balance; NGGB). NCB was defined as the sum of the annual total $CO_2$ and $CH_4$

492 exchanges. NGGB was defined in a similar fashion as the NCB, but considers the greater

493 radiative forcing potential of $CH_4$ than $CO_2$ (28 times over a 100-year horizon, Myhre et

494 al., 2013) when calculating the annual total. The calculated NCB values are mostly

495 negative because the stronger $CO_2$ uptake dominates the weaker $CH_4$ emissions (Figure

496 8c). The results suggest that all the examined peatland types serve as net carbon sinks

497 under current climate (CTRL), consistent with the estimates reported in Deng et al.

498 (2014) and Lundin et al. (2016). We find a 24, 36, and 38 g C $m^{-2}$ $y^{-1}$ underestimation of

499 NCB simulated in the palsa, bog, and fen sites, respectively, due to the cold and wet

500 biases in the GSWP3 climate reanalysis dataset (BIASED-COLD&BIASED-WET vs.

501 CTRL). NGGB is affected more strongly by $CH_4$ emissions (Figure 8d), due to its larger

502 radiative forcing potential. NGGB values are positive over the bog and fen sites,

503 suggesting that these sites can exhibit positive radiative forcing impacts despite being net

504 carbon sinks. NGGB simulated in the palsa site is generally negative (i.e., a net sink from

505 the atmosphere) due to lower $CH_4$ emissions, except for the simulation conducted without

506 any climate bias correction (correcting only air temperature increased $CH_4$ emissions but

507 not enough to compensate for the significantly higher $CO_2$ sink). Our results indicate that

508 the simulated NGGB would be biased by 298, -66, and -252 g $CO_2$-eq $m^{-2}$ $y^{-1}$ in the palsa,

509 bog, and fen sites, respectively, without proper bias correction for the GSWP3 climate

510 reanalysis dataset (BIASED-COLD&BIASED-WET vs. CTRL). Using the GSWP3



products directly thus effectively eliminates the positive radiative forcing from the
expanding bog and fen sites, while creating a potentially dramatically inaccurate positive
radiative forcing from the shrinking palsa sites.

**4.3 Climate sensitivity versus landscape heterogeneity**

Climate sensitivity and landscape heterogeneity are defined here as variability

across the four climate forcing datasets for a single peatland type, and variability across
three peatland types with bias-corrected climate (CTRL), respectively. We estimated
carbon cycle variability associated with climate sensitivity and landscape heterogeneity to
quantify the corresponding uncertainty in our annual carbon cycle assessments from 2003
to 2007. Our results indicate that differences in simulated annual mean $CO_2$ exchanges
and NCB from climate sensitivity are greater than those from landscape heterogeneity
(Figure 8a,c); i.e., annual $CO_2$ uptake strength is more sensitive to climate forcing
uncertainty than to peatland type representation. In terms of the simulated annual mean
$CH_4$ emissions and NGGB, our results indicate that variability from climate sensitivity is
comparable to those from landscape heterogeneity (Figure 8b,d). Therefore, bias-
corrected climate and realistic peatland characterization are both necessary to reduce the
uncertainty in representing $CH_4$ dynamics and its radiative forcing effects.

In addition to its effects on carbon cycle predictions, changes in climate

conditions also affect permafrost degradation and thus induce changes in areal cover of
peatland types. Malmer et al. (2005) showed that there were -0.95, 0.24, and 0.62 ha areal
cover changes (-10.3%, 4.0%, and 46.3% percentage changes) from 1970 to 2000 in
palsa, bog, and fen, respectively, at the Stordalen Mire. By applying the annual mean



$CO_2$ and $CH_4$ exchanges simulated with bias-corrected climate from 2003 to 2007, the
areal cover changes from 1970 to 2000 alone would lead to -44 kg C $y^{-1}$, 76 kg C $y^{-1}$, and
2076 kg $CO_2$-eq $y^{-1}$ changes in annual mean $CO_2$ exchanges, $CH_4$ exchanges, and NGGB,
respectively, at the Stordalen Mire. The changes in landscape scale carbon cycle
dynamics indicate that the radiative warming impact of increased $CH_4$ emissions is large
enough to offsets the radiative cooling impact of increased $CO_2$ uptake at the Stordalen
Mire, consistent with the estimates reported in Deng et al. (2014). The areal cover
changes across peatland types could persist or accelerate under the projected warming
and wetting trends in the Arctic (Collins et al., 2013; Bintanja and Andry, 2017), which
could stimulate $CH_4$ emissions and produce a stronger radiative warming impact.

**5. Conclusions**

We evaluated the climate bias in a widely used atmospheric reanalysis product

(GSWP3) at our northern Sweden Stordalen Mire site. We then applied a comprehensive
biogeochemistry model, *ecosys*, to estimate the effects of these biases on active layer
development and carbon cycling across a thaw gradient at the site. Our results show that
*ecosys* reasonably represented measured hydrological, thermal, and biogeochemical cycle
processes in the intact permafrost palsa, partly thawed bog, and fully thawed fen. We
found that the cold and wet biases in the GSWP3 climate reanalysis dataset significantly
alter model simulations, leading to biases in simulated Active Layer Depths, Net Carbon
Balance, and Net Greenhouse Gas Balance by up to 28.6%, 38 g C $m^{-2}$ $y^{-1}$, and 298 g
$CO_2$-eq $m^{-2}$ $y^{-1}$, respectively. The Net Carbon Balance simulated with bias-corrected
climate suggests that all the examined peatland types are currently net carbon sinks from





the atmosphere, although the bog and fen sites can have positive radiative forcing impacts
due to their higher $CH_4$ emissions.
Our results indicate that the annual means of ALD, $CO_2$ uptake, and $CH_4$
emissions generally increase along the permafrost thaw gradient at the Stordalen Mire
under current climate, consistent with previous studies in this region. Our analysis
suggests that palsa, bog, and fen landscape features differ strongly in their carbon cycling
dynamics and have different responses to climate forcing biases. Differences in simulated
$CO_2$ and $CH_4$ exchanges driven by uncertainty from climate forcing are as large as those
from landscape heterogeneity across the examined permafrost thaw gradient. Model
simulations demonstrate that the palsa site exhibits the strongest sensitivity to biases in
air temperature and precipitation. The wet bias in GSWP3 could erroneously increase
predicted $CH_4$ emissions from the palsa site to a magnitude comparable to emissions
currently measured in bog and fen sites. These results also show that increased
precipitation projected for high latitude regions could strongly accelerate $CH_4$ emissions
from the palsa area, even without degradation of palsa into bog and fen. Future studies
should thus recognize the effects of climate forcing uncertainty on carbon cycling, in
addition to tracking changes in carbon budget associated with areal changes in permafrost
degradation.

**Acknowledgements**
This study was funded by the Genomic Science Program of the United States Department
of Energy Office of Biological and Environmental Research under the ISOGENIE
project, grant DE-SC0016440, to Lawrence Berkeley Laboratory under contract DE-





AC02-05CH11231, and by support from the Swedish Research Council (VR) to PMC.
We thank the Abisko Scientific Research Station of the Swedish Polar Research
Secretariat for providing the meteorological data.





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





Table 1. Temporal coverage of quality-controlled $CO_2$ and $CH_4$ exchanges measured by
automatic chambers at the three peatland types in the Stordalen Mire during the years
2002 to 2007.

| | | $CO_2$ | | | $CH_4$ | |
|---|---|---|---|---|---|---|
| Sites | Number of data points | 3 Hourly coverage (%) | Daily coverage (%) | Number of data points | 3 Hourly coverage (%) | Daily coverage (%) |
| Palsa | 12752 | 65.8 | 12.4 | N/A | N/A | N/A |
| Bog | 12821 | 68.5 | 12.7 | 6660 | 96.2 | 25.0 |
| Fen | 8989 | 63.8 | 13.7 | 4923 | 90.5 | 33.7 |






Table 2. The evaluation of the 3 hourly and daily $CO_2$ and $CH_4$ exchanges simulated at
the palsa, bog, and fen sites. RRMSEs are relative root mean squared errors.

| | | 3-Hourly | | Daily | |
|---|---|---|---|---|---|
| | | $R^2$ | RRMSEs (%) | $R^2$ | RRMSEs (%) |
| Sites | C component | | | | |
| Palsa | $CO_2$ | 0.48 | 13.4 | 0.36 | 18.3 |
| Bog | $CO_2$ | 0.63 | 19.1 | 0.44 | 35.8 |
| | $CH_4$ | 0.31 | 16.3 | 0.47 | 22.3 |
| Fen | $CO_2$ | 0.64 | 8.4 | 0.43 | 25.5 |
| | $CH_4$ | 0.44 | 11.1 | 0.54 | 16.9 |







Table 3. Means and standard deviations of cumulative $CO_2$ and $CH_4$ exchanges simulated
at the palsa, bog, and fen sites during the period 2003 to 2007. All units are represented in
g C m$^{-2}$.

| Sites | C flux component | Growing season; Days 119–288 | | Non-growing season; Days 1–118/289–365 | |
|---|---|---|---|---|---|
| | | Mean | Standard deviation | Mean | Standard deviation |
| Palsa | | | | | |
| | $CO_2$ | -72.70 | 19.10 | 38.89 | 4.09 |
| | $CH_4$ | 0.04 | 0.02 | 0.01 | 0.002 |
| Bog | | | | | |
| | $CO_2$ | -79.59 | 21.46 | 42.89 | 2.16 |
| | $CH_4$ | 3.52 | 0.45 | 0.42 | 0.11 |
| Fen | | | | | |
| | $CO_2$ | -88.65 | 7.26 | 44.41 | 6.13 |
| | $CH_4$ | 10.86 | 3.95 | 0.78 | 0.18 |







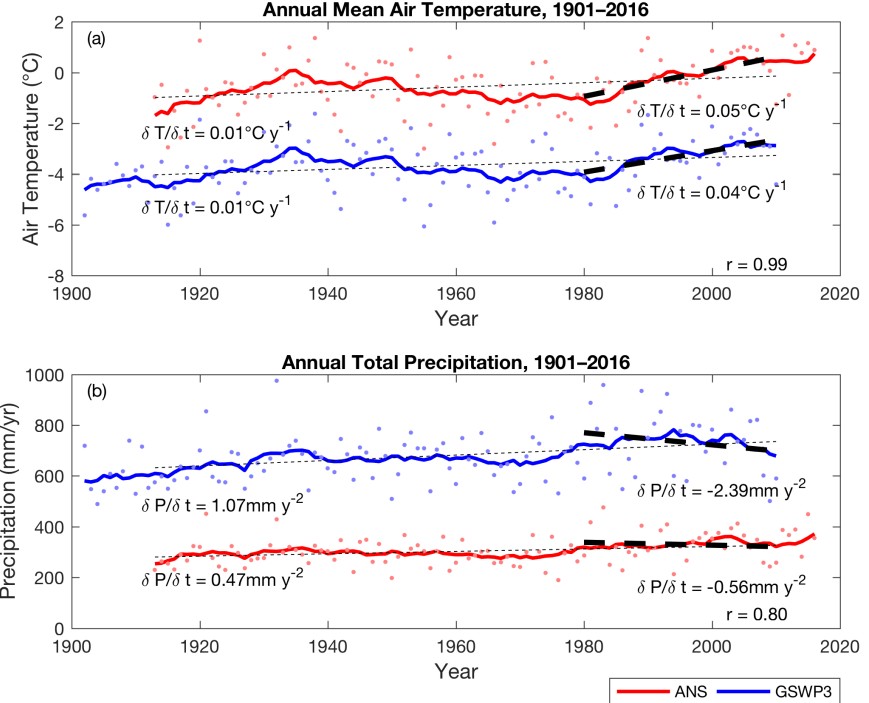


Figure 1. Time series of the air temperature (a) and precipitation (b) measured at ANS
(red; years 1913–2016) and extracted from GSWP3 (blue; years 1901–2010). Dots are
the annual means and solid lines are the decadal moving averages of the corresponding
annual means. Thin and thick dashed lines are the trends for years 1913–2010, and years
1980–2010, respectively. The inset r values are the correlation coefficients calculated
between the two time series.



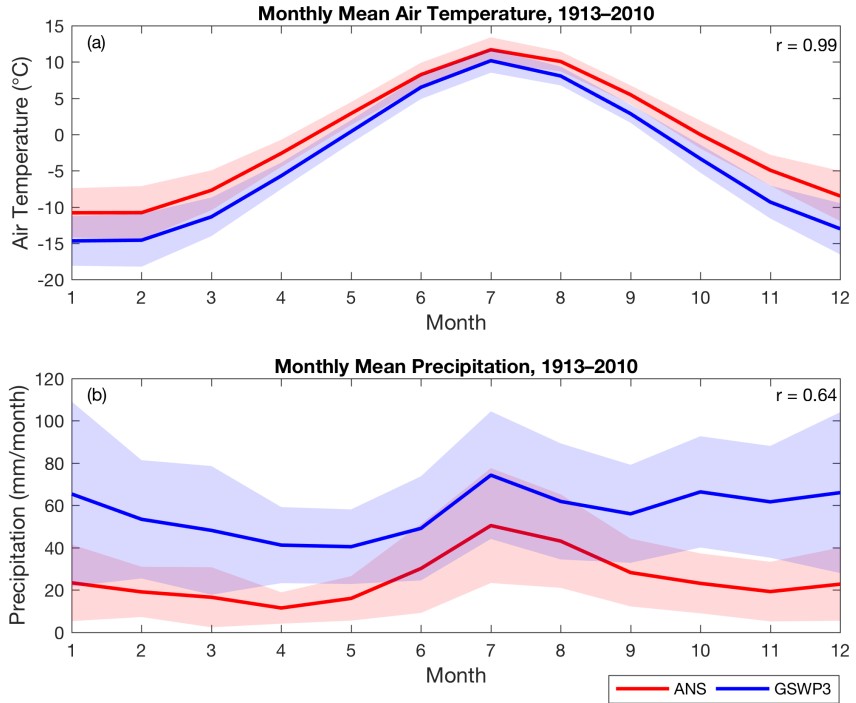


Figure 2. Monthly mean air temperature (a) and precipitation (b) measured at ANS (red)
and extracted from GSWP3 (blue). The shaded area is the inter-annual variability for the
corresponding dataset, represented by the standard deviations calculated at each month.
The inset r values are the correlation coefficients calculated between the two time series.



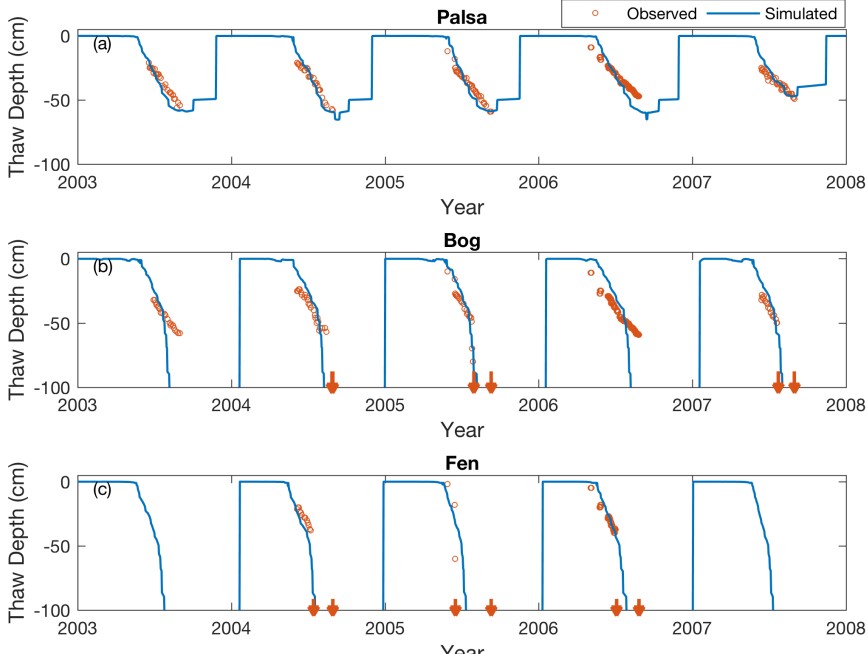


Figure 3. Simulated (solid lines) and measured (open circles) seasonal dynamics of thaw
depth at the palsa (a), bog (b), and fen (c) sites from 2003 to 2007. Downward arrows
indicate the time period that the measured thaw depth is deeper than 90 cm for a
measurement year.





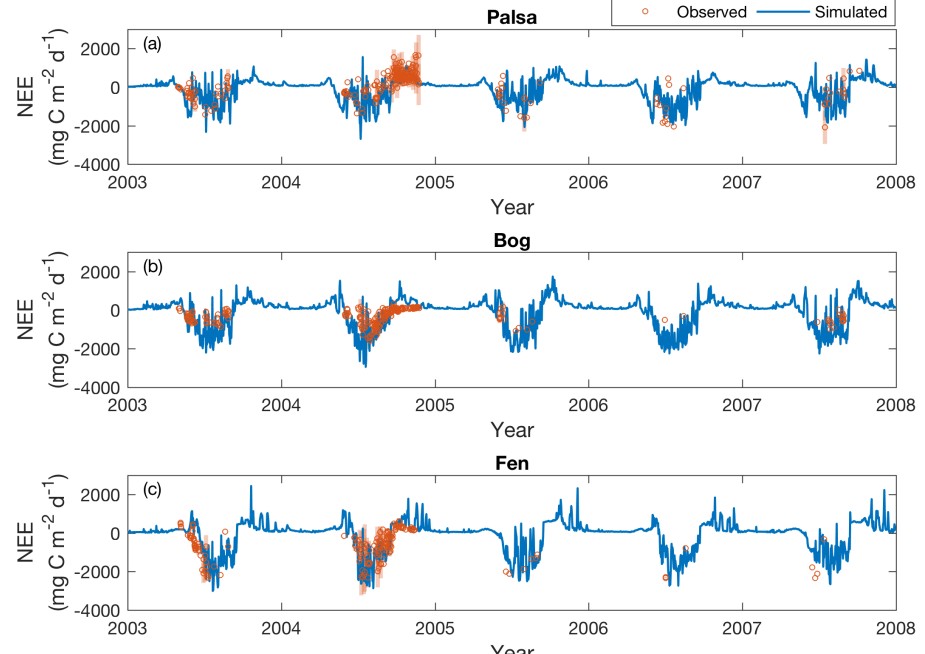


Figure 4. Simulated (solid lines) and measured (open circles) daily $CO_2$ exchanges (NEE)
at the palsa (a), bog (b), and fen (c) sites, from 2003 to 2007. Shaded bars are the
standard deviations of the daily NEE measured across the subsites under each peatland
type. The positive values indicate effluxes, and the negative values indicate influxes.





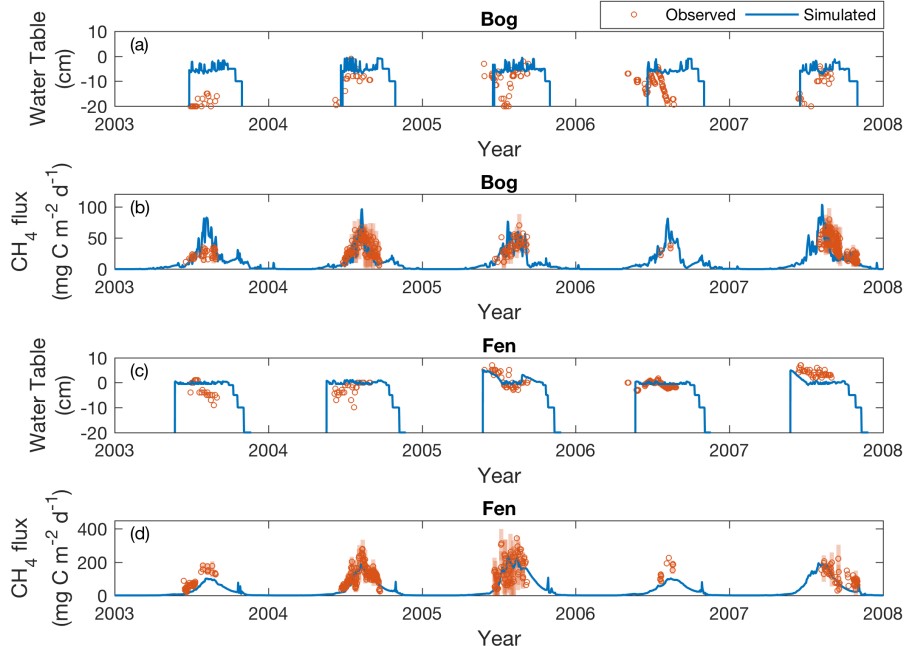


Figure 5. Simulated (solid lines) and measured (open circles) water table depth and daily
CH$_4$ emissions at the bog and fen sites from 2003 to 2007. Shaded bars are the standard
deviations of the daily CH$_4$ emissions measured across the subsites under each peatland
type.





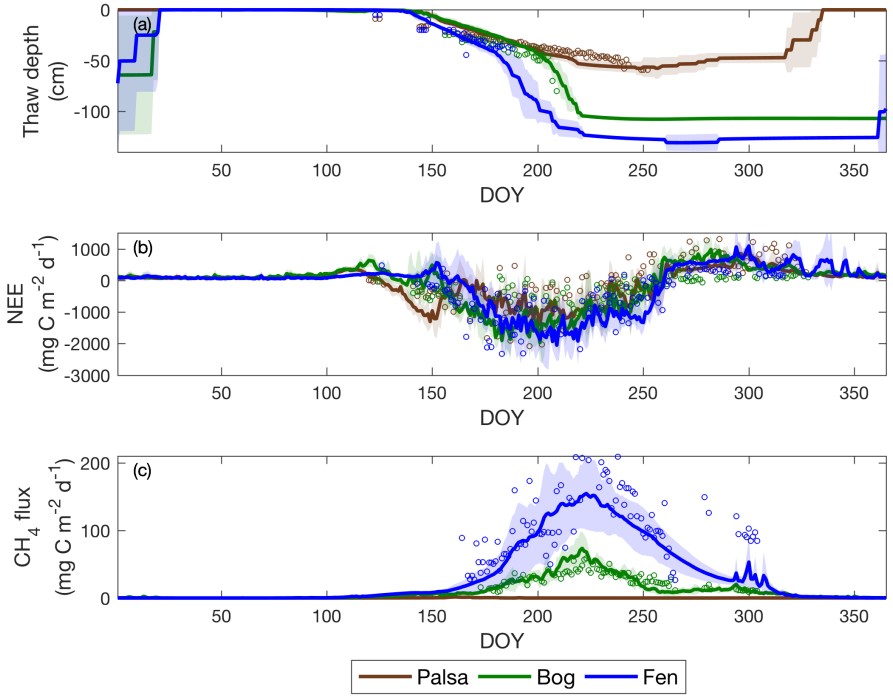


Figure 6. Daily composite results of (a) thaw depth, (b) daily NEE, and (c) daily $CH_4$

exchanges across the thaw gradient from 2003 to 2007. Solid lines and open circles are

the simulated and measured inter-annual means, respectively. The shaded area is the

simulated inter-annual variability for the corresponding dataset, represented by the

standard deviations calculated at each day of year. The positive values indicate effluxes,

and the negative values indicate influxes.






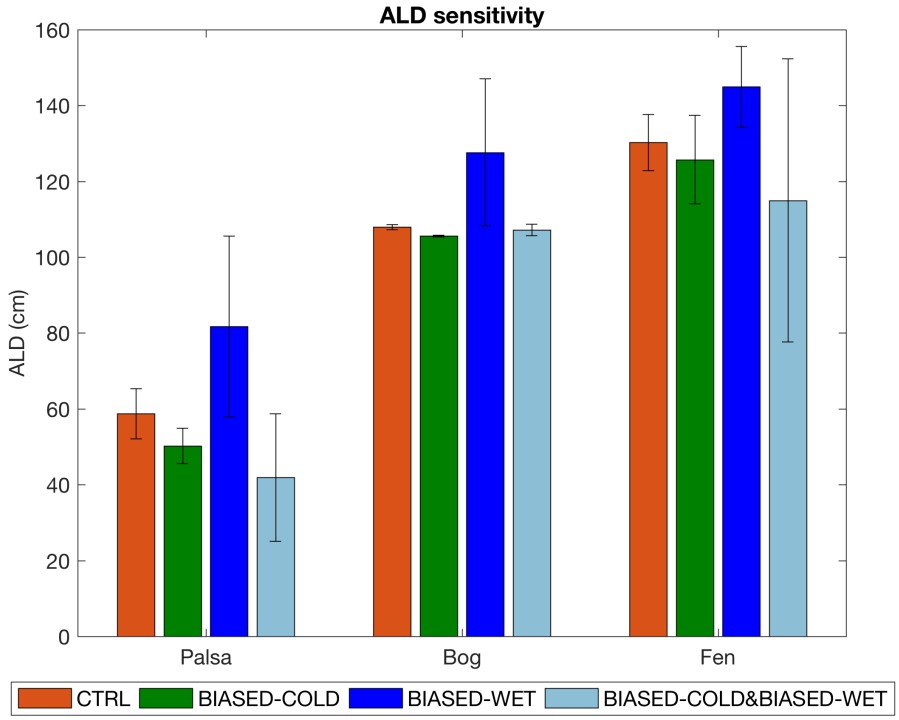


Figure 7. Simulated ALD at the palsa, bog, and fen sites, for four sets of climate forcing
(Section 2.5). Bars and error bars are the means and standard deviations calculated from
2003 to 2007, respectively.




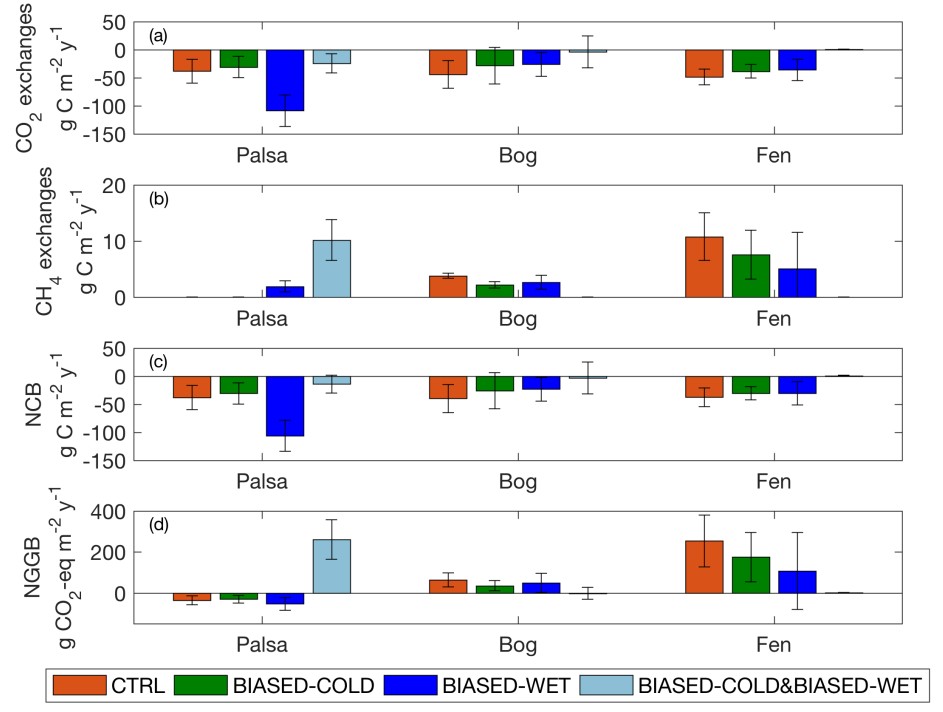


Figure 8. Annual $CO_2$ exchanges (a), $CH_4$ exchanges (b), Net Carbon Balance (c), and

Net Greenhouse Gas Balance (d) simulated at the palsa, bog, and fen sites, under each set

of simulations. Bars and error bars are the means and standard deviations calculated from

2003 to 2007, respectively. The positive values indicate effluxes, and the negative values

indicate influxes.

961

962