# Peer review of "Large carbon cycle sensitivities to climate across a permafrost thaw gradient in subarctic"

_The Cryosphere, 2018_

## Referee Comment (RC1) · Anonymous Referee #1 · 6 Nov 2018

This study used a biogeochemical model to examine effects of climate forcing biases in global climate reanalysis on carbon cycle predictions across a permafrost peatland thaw gradient. The main findings show that all peatland sites studied (bog, fen, palsa) remain carbon sinks, but that the bog and fen have a net positive radiative forcing because of high methane emissions. The study finds that climate responses can have major implications for carbon cycle dynamics in these systems. It is well written. Just a few comments below to help clarify some of the site descriptions, etc.

Specific comments:

Line 63-69: what about the mass-balance studies from Alaska that suggest that a

significant portion of the permafrost peat is lost upon thaw (O'Donnell et al., 2014 Ecosystems, Jones et al., 2016 Global Change Biology).

Line 135-136: The seasonality of precipitation could be important. Is there information on whether its increased snowfall/depth (which could also warm soil temperatures)?

Study site description: has anyone studied the history of permafrost at this site (i.e., when it formed) and whether permafrost aggradation occurred syngenetically with peat accumulation? I would argue that this information is important in carbon dynamics with thaw.

Line 161: italicize Sphagnum

Line 173: It is unclear if the measurements described in this paragraph were conducted in this study or if the authors are reporting on measurements made by Bäckstrand et al. Perhaps that can be clarified at the beginning of this paragraph

Line 174-175: "chamber lids were removed in the Fall": Fall can be lowercase (as can spring, here and elsewhere; you don't capitalize "summer" later in the text). Can you be more specific about "fall" and "spring"? How closely did the measurements coincide with freeze-up and thaw? Which months? Do you suspect that there are winter emissions? How many chambers/peatland type? Do you have any idea when the fen and bog thawed (i.e., 5 years ago, 500 years ago, 1000 years ago) and when peat started accumulating at these sites? These could have important implications for emissions and the carbon balance of a peatland.

Line 365: how much does the water table fluctuate for the bog over the summer?

Line 408: "the higher CH4 emissions in the fully thawed fen can be attributed to its faster thaw rate": Do you mean rate of seasonal thaw or do you mean rate of permafrost thaw? If permafrost thaw, how do you know how quickly it thawed?

Section 4.1, ∼line 415: in addition to ALD, do you know if there is a talik in any of these peatland types in the winter? Also, I don't understand why the "fully thawed fen" or the

bog have an ALD, if they're thawed. Perhaps a conceptual diagram would help readers envision the differences in permafrost regime of these different peatland types, or at least clarification about what is meant by ALD in the "fully thawed fen" and bog. An additional table might be useful that includes information about total peat depth, active layer depths, average water table depths, surface vegetation communities, and perhaps some information on the number of chambers per site (and was just one feature per peatland type studied or did you study multiple?)?

Line 481-482: Is the model dynamic? Is vegetation allowed to change as conditions change (wetter, drier), or did the model not run long enough for species changes to occur?

Please also note the supplement to this comment:
https://www.the-cryosphere-discuss.net/tc-2018-215/tc-2018-215-RC1-supplement.pdf

---

## Referee Comment (RC2) · Anonymous Referee #2 · 11 Nov 2018

General comments: This study applied the ecosys model to predict soil thaw dynamics, NEE, and CH4 fluxes across a permafrost thaw gradient encompassing palsa, bog, and fen at a subarctic peatland. The authors also investigated impacts of potential climate bias on the simulated active layer depth (ALD), NEE, and CH4 fluxes. My major concern is that this manuscript is lacking of clearness for methods and explanation/discussion for some results. While the authors cited a lot of ecosys related references, it is not clear how this model simulate ALD, water table, NEE, and CH4 fluxes (i.e. lacking of the model structure/principle/processes related to these variables) and how the authors set different parameters for palsa, bog, and fen to simulate these variables for different land types (i.e. lacking of description for setting of some parameters;

[Figure]

please check specific points). I therefore suggest adding these contents. In addition, I noticed some indigestible results, but did not find explanations/discussions related to these results (check specific comments). I suggest adding explanations/discussions for these results. Specific comments: Line 151: ALD should be defined at the first place of 'active layer depth'. Line 151: For clarity, I suggest changing '35 cm' to '35 cm below the peat surface'. Lines 240 to 241: Did you mean that you used the climate data from 1901 to 2010 for model initialization? If so, which year's results were used for analysis? Line 268: How about the values for bog? Lines 264 to 274: How about the vegetation parameters for these three land types? Lines 296 to 298: In this sentence, did you want to say inter-annual variability of GSWP3 temperature is smaller in summer. If so, I suggest adding information of the inter-annual variability, in addition to information of underestimation. Lines 336 to 337: This is not accurate; I noticed some points of net $CO_2$ emission during summer from the figure 4. Line 345: What is the meaning of 'different subsites' in this and other places? Different chambers for a peatland type? Lines 351 to 356: I noticed consistent over-predictions of net $CO_2$ uptake for bog. Could you please provide some explanations? Lines 372 to 375: It is not clear for me how the authors simulated different water table for different land types without considering lateral water transport. Was the simulated different WT driven by different ET among the types since they had the same rainfall? In addition, why this is 'a particular issue' for bog considering that fen receives a large amount of water from a lack (Line 153)? Line 390: The 'weaker CH4 emission variability measured across subsites' is confusing. Line 399: I cannot catch this sentence. The model can produce hourly/daily results, so it is easily to calculate seasonal cumulative NEE directly using the simulations. Why you calculate it based on the seasonality identified in another paper? Lines 427 to 429: Could you please explain why the simulated ALDs at palsa and fen under cold and wet conditions are shallower than that under cold conditions? This seems not consistent with the comparisons between wet and control. Lines 451 to 452: Why the fen showed weak $CO_2$ emissions under cold and wet conditions and net $CO_2$ uptake under cold conditions, due to reduced NPP and/or increased soil respiration under wetter conditions? I cannot understand the large impacts of wet on CO2 emissions at fen given that WT is close to or above ground surface for this site (Figure 5). Lines 483 to 487: Could you explain the simulated negative impacts of wet on CH4 emissions at bog and fen; in particular for the cold and wet scenario, why the CH4 emissions was simulated close to zero?

---

## Author Comment (AC1) · 10 Jan 2019

The authors would like to thank the Anonymous Referee 1 for his/her valuable comments and suggestions to strengthen the analysis presented in our manuscript. The comments and suggestions have been taken into account in the revised manuscript, as follows (original referee's comments in bold):

This study used a biogeochemical model to examine effects of climate forcing biases in global climate reanalysis on carbon cycle predictions across a permafrost peatland thaw gradient. The main findings show that all peatland sites studied (bog, fen, palsa) remain carbon sinks, but that the bog and fen have a net positive radiative forcing

[Figure]

because of high methane emissions. The study finds that climate responses can have major implications for carbon cycle dynamics in these systems. It is well written. Just a few comments below to help clarify some of the site descriptions, etc.

The authors thank the reviewer for the valuable comments. We have carefully revised our manuscript based on the suggestions provided by the reviewer to clarify some of the original descriptions.

Specific comments:

Line 63-69: what about the mass-balance studies from Alaska that suggest that a significant portion of the permafrost peat is lost upon thaw (O'Donnell et al., 2014 Ecosystems, Jones et al., 2016 Global Change Biology).

O'Donnell et al. (2012) and Jones et al. (2017) have been included in our revised manuscript (Lines 66-71) to improve our description of permafrost peatland carbon vulnerability under a changing climate.

Line 135-136: The seasonality of precipitation could be important. Is there information on whether its increased snowfall/depth (which could also warm soil temperatures)?

The precipitation measured at ANS did not show a significant change in precipitation seasonality throughout the years. Precipitation magnitude and variability during summer are generally greater than during winter (Figure 3b). The analysis presented by Malmer et al. (2005) showed that monthly mean snow depth in 1986-2000 was greater than in 1957-1970, suggesting that it is possible that increased annual precipitation (Figure 2b) can be associated with increased snowfall in winter and spring. Unfortunately, we do not have continuous snowfall or snow depth measurements to compare with the changes in precipitation.

Study site description: has anyone studied the history of permafrost at this site (i.e., when it formed) and whether permafrost aggradation occurred syngenetically with peat accumulation? I would argue that this information is important in carbon dynamics with

thaw.

Inception of peat deposition at the Stordalen Mire has been dated at around 6,000 calendar years before present (cal. BP) (Sonesson 1972) in the southern part of the mire and at around 4,700 cal. BP in the northern part (Kokfelt et al., 2010). Kokfelt et al. (2010) suggested that permafrost aggregation initiated during the Little Ice Age (around 120-400 cal. BP) in the Stordalen Mire. We agree with the reviewer that permafrost history of the study site could be important in carbon dynamics with thaw. However, such information may not affect our analysis because we did not attempt to simulate permafrost aggradation/degradation history in this study. Instead, we simulate the palsa, bog, and fen sites individually and discuss the differences shown in different thaw stages. We did, however, add the peat and permafrost history information to the site description (Lines 146-150).

Line 161: italicize Sphagnum

Sphagnum has been italicized as suggested by the reviewer (Line 171).

Line 173: It is unclear if the measurements described in this paragraph were conducted in this study or if the authors are reporting on measurements made by Bäckstrand et al. Perhaps that can be clarified at the beginning of this paragraph

The measurements described in this paragraph were made by BaÌĹckstrand et al. (2008b). A proper citation has been added at the beginning of this paragraph (Line 184).

Line 174-175: "chamber lids were removed in the Fall": Fall can be lowercase (as can spring, here and elsewhere; you don't capitalize "summer" later in the text). Can you be more specific about "fall" and "spring"? How closely did the measurements coincide with freeze-up and thaw? Which months? Do you suspect that there are winter emissions? How many chambers/peatland type? Do you have any idea when the fen and bog thawed (i.e., 5 years ago, 500 years ago, 1000 years ago) and when

peat started accumulating at these sites? These could have important implications for emissions and the carbon balance of a peatland.

All seasons have been converted to lowercase in the revised manuscript. We have revised the original sentence with some clarifications for the measurement period (Lines184-188). We believe that winter emissions could contribute to the annual carbon budget, as suggested by our simulation results, but we don't yet have enough quality-controlled measurements in winter to verify our hypothesis. We don't yet know precisely when the fen and bog formed, but all three of the investigated peatland types were there before 1930's (based on Swedish military photography; information added to the site description (Lines 164-165)). As mentioned above, Kokfelt et al. (2010) suggested that peat inception took place at around 4,700 cal. BP (around 6,000 cal. BP by Sonesson 1972) in the Stordalen Mire and permafrost aggregation initiated during the Little Ice Age.

Line 365: how much does the water table fluctuate for the bog over the summer?

The simulated water table fluctuates from -7 cm to -1 cm (below surface) over the summer, as shown in Figure 6.

Line 408: "the higher $CH_4$ emissions in the fully thawed fen can be attributed to its faster thaw rate": Do you mean rate of seasonal thaw or do you mean rate of permafrost thaw? If permafrost thaw, how do you know how quickly it thawed?

We meant the rate of seasonal thaw, not the rate of permafrost thaw. We simulated the seasonal thaw dynamics under three different permafrost thaw stages, and compared the seasonal thaw and carbon cycling across the permafrost thaw gradient. The original sentence has been revised as suggested by the reviewer (Line 421).

Section 4.1, âĹijline 415: in addition to ALD, do you know if there is a talik in any of these peatland types in the winter? Also, I don't understand why the "fully thawed fen" or the bog have an ALD, if they're thawed. Perhaps a conceptual diagram would help

readers envision the differences in permafrost regime of these different peatland types, or at least clarification about what is meant by ALD in the "fully thawed fen" and bog. An additional table might be useful that includes information about total peat depth, active layer depths, average water table depths, surface vegetation communities, and perhaps some information on the number of chambers per site (and was just one feature per peatland type studied or did you study multiple?)?

The water table depth and ALD were only measured in the growing season, and we do not know if there is a talik in our study sites during winter. The confusing description (fully thawed fen) has been replaced with "fen" in the revised manuscript. The original terminology was chosen to match previous studies conducted at the same sites that qualitatively describes the permafrost thaw gradient across palsa, bog, and fen. As suggested by the reviewer, an additional conceptual diagram (Figure 1) has been included in the revised manuscript to provide a qualitative summary of the three peatland types investigated in this study.

Line 481-482: Is the model dynamic? Is vegetation allowed to change as conditions change (wetter, drier), or did the model not run long enough for species changes to occur?

Vegetation is allowed to change with changing environmental conditions, and we noticed species changes among different model forcing climate conditions (results not shown). The ecosys model prognoses vegetation dynamics with internal resource allocation and remobilization, competition for light and nutrients, and different plant functional traits. Shifts in plant functional types were modeled through processes of plant functional type competition for light, water, and nutrients (nitrogen, and phosphorus) within each canopy and rooted soil layer. A qualitative summary of the ecosys model has been included in the supplemental material of this manuscript.

Reference

Kokfelt, U., Reuss, N., Struyf, E., Sonesson, M., Rundgren, M., Skog, G., Rosen, P.,

and Hammarlund, D.: Wetland development, permafrost history and nutrient cycling inferred from late Holocene peat and lake sediment records in subarctic Sweden, J. Paleolimn., 44, 327–342, doi:10.1007/s10933-010-9406- 8, 2010.

Malmer, N., Johansson, T., Olsrud, M. and Christensen, T. R.: Vegetation, climatic changes and net carbon sequestration in a North-Scandinavian subarctic mire over 30 years, Global Change Biology, 11(11), 1895–1909, doi:10.1111/j.1365-2486.2005.01042.x, 2005. Oksanen PO (2006) Holocene development of the vaisjea ÌĹggi palsa mire, finnish lapland. Boreas 35:81–95.

Sonesson M (1972) Cryptogams. In: International biological programme—Swedish tundra biome project. Technical report No. 9, April 1972. Swedish Natural Science Research Council Ecological Research Committee

Zuidhoff F, Kolstrup E (2000) Changes in palsa distribution in relation to climate change in Laivadalen, northern Sweden, especially 1960–1997. Permafrost Periglac 11:55–59

Please also note the supplement to this comment:
https://www.the-cryosphere-discuss.net/tc-2018-215/tc-2018-215-AC1-supplement.pdf
* * *

---

## Author Comment (AC2) · 10 Jan 2019

The authors would like to thank the Anonymous Referee 2 for his/her valuable comments and suggestions to strengthen the analysis presented in our manuscript. The comments and suggestions have been taken into account in the revised manuscript, as follows (original referee's comments in bold):

General comments:

This study applied the ecosys model to predict soil thaw dynamics, NEE, and CH4 fluxes across a permafrost thaw gradient encompassing palsa, bog, and fen at a subarctic peatland. The authors also investigated impacts of potential climate bias on the simulated active layer depth (ALD), NEE, and CH4 fluxes. My major concern is that this manuscript is lacking of clearness for methods and explanation/discussion for some results. While the authors cited a lot of ecosys related references, it is not clear how this model simulate ALD, water table, NEE, and CH4 fluxes (i.e. lacking of the model structure/principle/processes related to these variables) and how the authors set different parameters for palsa, bog, and fen to simulate these variables for different land types (i.e. lacking of description for setting of some parameters; please check specific points). I therefore suggest adding these contents. In addition, I noticed some indigestible results, but did not find explanations/discussions related to these results (check specific comments). I suggest adding explanations/discussions for these results.

The authors thank the reviewer for the valuable comments, which we believe have improved the manuscript substantially. We have included a qualitative summary of the ecosys model to improve our model structure description. We have included the list of parameters we used for our simulation to improve our parameter description. We have included the necessary explanations and discussions requested by the reviewer to improve the clarity of our manuscript.

Specific comments:

Line 151: ALD should be defined at the first place of 'active layer depth'.

We have now defined ALD as 'active layer depth' at the first place of its presence (Line 108).

Line 151: For clarity, I suggest changing '35 cm' to '35 cm below the peat surface'.

We have applied the wording suggested by the reviewer to improve clarity (Line 160).

Lines 240 to 241: Did you mean that you used the climate data from 1901 to 2010 for model initialization? If so, which year's results were used for analysis?

The climate data from 1901 to 2001 were used for model initialization (i.e., spinup) and those from 2002 to 2010 were used for analysis. We have clarified this approach in the revised Methods section (Lines 254-255)

Line 268: How about the values for bog?

RydeÌẠn et al. (1980) did not specify the differences between bog and fen, so we applied the same soil bulk density values for the upper part of the bog and fen sites. We have included a table (Supplemental Material Table 1) to specify some of the key model parameters used in our simulation.

Lines 264 to 274: How about the vegetation parameters for these three land types?

Vegetation parameters at the three peatland types were assigned by the observed plant species (section 2.1; Figure 1). The palsa, bog, and fen sites were composed by 4 (shrubs, mosses, sedges, and lichens), 2 (mosses and sedges), and 1 (sedges) plant functional types, respectively.

Lines 296 to 298: In this sentence, did you want to say inter-annual variability of GSWP3 temperature is smaller in summer. If so, I suggest adding information of the inter-annual variability, in addition to information of underestimation.

The inter-annual variability information has been added to this sentence (Lines 314-316).

Lines 336 to 337: This is not accurate; I noticed some points of net $CO_2$ emission during summer from the figure 4.

The sentence has been revised to account for the $CO_2$ emission events during summer (Line 355).

Line 345: What is the meaning of 'different subsites' in this and other places? Different chambers for a peatland type?

'Different subsites' was used to indicate different automated chambers for a given peatland type. We have annotated the definition of subsites to improve the clarity of our manuscript (Lines 189 and 363).

Lines 351 to 356: I noticed consistent over-predictions of net CO2 uptake for bog. Could you please provide some explanations?

The authors agree with the reviewer that the simulated net CO2 uptake, indicated as negative NEE, were sometimes greater than the measured values during summer. The over-predictions of net CO2 uptake for the bog could be due to overestimated plant biomass or overestimated CO2 uptake rate per biomass. However, we currently don't have data to examine the actual cause of overestimated net CO2 uptake for the bog since the CO2 flux derived from automated chambers only represents the aggregated results of all controlling factors. An additional dataset of plant biomass (for mosses and sedges at individual automated chamber locations) is needed to examine the cause of overestimated net CO2 uptake for bog.

Lines 372 to 375: It is not clear for me how the authors simulated different water table for different land types without considering lateral water transport. Was the simulated different WT driven by different ET among the types since they had the same rainfall? In addition, why this is 'a particular issue' for bog considering that fen receives a large amount of water from a lack (Line 153)?

The WTD in ecosys is calculated at the end of each time step as the depth to the top of the saturated zone below which air-filled porosity is zero. Changes in the simulated water table (WT) were driven by dynamical interactions among precipitation, ET, vertical water transport, and lateral water transport. An external WT was prescribed in our simulations, and that WT interacts with our one-dimensional gridcell. However, the one-dimensional simulation cannot account for lateral water transport among landscape features within a system. For example, no additional water could be transported from the neighboring grids to lift local WT, and no excessive water could be transported to the neighboring grids to deepen local WT. We believe that such processes could be

important in determining local WT, but those effects could not be represented in our one-dimensional column simulation. Although both bog and fen WTs were affected by the limitations of our one-dimensional column simulation, it could be more of a concern for bog because the measured WT variability is stronger in bog than those in fen.

Line 390: The 'weaker CH4 emission variability measured across subsites' is confusing.

This sentence was meant to indicate that the variability of CH4 emissions measured across subsites (automated chambers) within a given peatland type was weaker than CO2 variability, so increased temporal resolution (helpful for reducing variability across subsites) did not improve our evaluation of CH4 emissions. This sentence has been revised to improve its clarity (Lines 408-410).

Line 399: I cannot catch this sentence. The model can produce hourly/daily results, so it is easily to calculate seasonal cumulative NEE directly using the simulations. Why you calculate it based on the seasonality identified in another paper?

We calculated seasonal cumulative NEE directly using our simulation results, but with the green and snow seasonality instead of the quarterly seasonality. We chose to apply the green and snow seasonality identified in Bäckstrand et al. (2010) to help facilitate the inter-comparison of carbon budgets estimated in the Stordalen Mire, and to better capture the actual seasonality recorded at the study site.

Lines 427 to 429: Could you please explain why the simulated ALDs at palsa and fen under cold and wet conditions are shallower than that under cold conditions? This seems not consistent with the comparisons between wet and control.

The simulated ALDs in BIASED-WET were deeper than those in CTRL because the increased snowpack depth keeps the soil warmer with lower soil ice content during winter. A similar snowpack warming mechanism was found in the comparisons between BIASED-COLD and BIASED-COLD&BIASED-WET (i.e., soil ice content was

lower with the additional snowpack from the wet biases); however, summertime soil heating in some of the simulation years was not strong enough to thaw the soil ice between 20-40 cm completely with the cold biases. The presence of ice in the middle of the soil column in the BIASED-COLD&BIASED-WET run thus reduces the simulated ALD in some of the simulation years and results in shallower mean ALD as compared to the BIASED-COLD run. These descriptions have now been added to the revised manuscript (Lines 449-452).

Lines 451 to 452: Why the fen showed weak $CO_2$ emissions under cold and wet conditions and net $CO_2$ uptake under cold conditions, due to reduced NPP and/or increased soil respiration under wetter conditions? I cannot understand the large impacts of wet on $CO_2$ emissions at fen given that WT is close to or above ground surface for this site (Figure 5).

The vegetation structure and function simulated in ecosys dynamically respond to changes in environmental conditions. The amount of sedges simulated in the fen becomes lower under colder/wetter environment (BIASED-COLD/ BIASED-WET vs. CTRL), which slightly weakens the simulated $CO_2$ uptake strength (Figure 9a). When cold and wet biases are coupled together, simulated $CO_2$ uptake in the fen was substantially reduced in the BIASED-COLD&BIASED-WET run due to increased oxygen stress. Therefore, the simulated GPP/NPP is significantly reduced in the BIASED-COLD&BIASED-WET run, which shifts the fen toward a weak source of $CO_2$ emissions (Figure 9a). These descriptions have now been added to the revised manuscript (Lines 474-475).

Lines 483 to 487: Could you explain the simulated negative impacts of wet on $CH_4$ emissions at bog and fen; in particular for the cold and wet scenario, why the $CH_4$ emissions was simulated close to zero?

As described above, the presence of wet biases (BIASED-COLD and BIASED-COLD&BIASED-WET) reduces oxygen exchange, which reduces heterotrophic

respiration, microbial biomass, and the amount of CH4 production. The reduction of sedges under wetter environment (BIASED-COLD and BIASED-COLD&BIASED-WET) weakens aerenchyma transport, which also limits CH4 emissions. When cold and wet biases are coupled together, both of these effects (reduced CH4 production and weaker aerenchyma transport) strongly inhibit CH4 emissions and greatly reduce the simulated CH4 exchanges in the bog and fen sites. These descriptions have now been added to the revised manuscript (Lines 509-512).

Please also note the supplement to this comment:
https://www.the-cryosphere-discuss.net/tc-2018-215/tc-2018-215-AC2-supplement.pdf

---

## Author Comment (AC3) · 10 Jan 2019

**1** Model description**

2 *Ecosys* represents multiple canopy and soil layers and fully coupled carbon, energy, 3 water, and nutrient cycles solved at an hourly time step. Surface energy and water exchanges 4 drive soil heat and water transfers to determine soil temperatures and water contents. These 5 transfers drive soil freezing and thawing and, hence, active layer depth, through the general heat 6 flux equation. Carbon uptake is controlled by plant water status calculated from convergence 7 solutions that equilibrate total root water uptake with transpiration. Atmospheric warming 8 increases surface heat advection, soil heat transfers, and hence active layer depth. Canopy 9 temperatures affect CO2 fixation rates from their effects on carboxylation and oxygenation 10 modeled with Arrhenius functions for light and dark reactions. Soil temperatures affect 11 heterotrophic respiration through the same Arrhenius function as for dark reactions. 12 Carbon uptake is also affected by plant nitrogen uptake. The model represents fully coupled transformations of soil carbon, nitrogen, and phosphorus through microbially driven 13 14 processes. Soil warming enhances carbon uptake by hastening microbial mineralization and root nitrogen uptake. Carbon uptake is affected by phenology with leafout and leafoff (deciduous 15 plants) or dehardening and hardening (evergreen plants) being determined by accumulated 16 17 exposure to temperatures above set values while day length is increasing or below set values while day length is decreasing. Senescence is driven by excess maintenance respiration and by 18

20

19

**21 *Ecosystem-Atmosphere energy exchange:**

phenology in deciduous plant functional types.

Canopy energy and water exchanges in *ecosys* are calculated through a multi-layered
 soil-root-canopy system. The clumping effect for each leaf and stem surface is represented by a

24 species-specific interception fraction to simulate non-uniformity in the horizontal distribution of 25 leaves within each canopy layer. Coupled first-order closure schemes are solved between the atmosphere and each of leaf and stem surfaces in the multi-layered canopy to achieve energy 26 27 balance at each model time step. Once the system converges to the required canopy temperature, 28 latent and sensible heat fluxes of each canopy layer are calculated based on the simulated vapor 29 pressure deficit, canopy-atmosphere temperature gradient, aerodynamic conductance, and 30 stomatal conductance. Canopy heat storage is calculated from changes in canopy temperature and heat capacities of leaves, twigs, and stems. 31

32

33 *Canopy water relations:*

A convergence solution is sought for the canopy water potential of each plant population at which the difference between its transpiration and total root water uptake equals the difference between its water contents at the previous and current water potentials. Canopy water potential controls transpiration and soil-root water uptake, which affects stomatal conductance and thereby all the processes (e.g., canopy temperature and vapor pressure) described in "Ecosystem-Atmosphere energy exchange".

40

41 *Canopy carbon and nutrient cycling:*

Leaf carboxylation rates are adjusted from those calculated under non-limiting water potential to those under current water potential. The gross canopy CO2 fixation is the sum of the leaf carboxylation rate of each leaf surface present on each branch of each plant species, which is then transported to a mobile pool of carbon storage. Storage carbon oxidized in excess of maintenance respiration requirements is used as growth respiration to drive the formation of new

biomass. Net CO2 fixation is calculated as the difference between gross fixation and the sum of
maintenance, growth, and senescence respiration in the simulated canopy.

49 Nutrient (nitrogen and phosphorous) uptake is calculated for each plant species by 50 solving for aqueous concentrations at root and mycorrhizal surfaces in each soil layer at which 51 radial transport by mass flow and diffusion from the soil solution to the surfaces equals active 52 uptake by the surfaces. This solution dynamically links rates of soil nutrient transformations with 53 those of root and mycorrhizal nutrient uptake. The products of nitrogen and phosphorous uptake 54 are transported to mobile pools of nitrogen and phosphorous stored in each root and mycorrhizal 55 layer, which regulate vegetation growth.

56

**57 *Plant functional type dynamics:**

58 The model represents prognostic vegetation dynamics with internal resource allocation and remobilization. Shifts in plant functional types are modeled through processes of plant 59 functional type competition for light, water, and nutrients within each canopy and rooted soil 60 61 layer depending on leaf area and root length. Each plant functional type competes for nutrient 62 and water uptake from common nutrient and water stocks held across multi-layer soil profiles, calculated from algorithms for transformations and transfers of soil carbon, nitrogen, and 63 phosphorus, and for transfers of soil water. Modeled differences in plant functional type 64 functional traits determine the strategy of resource acquisition and allocation that drive growth, 65 66 resource remobilization, and litterfall, and therefore each plant functional type dynamic competitive capacity under different environmental conditions. 67

68

69 *Soil microbial activity:*

70 The modeling of microbial activity is based on six organic states: solid, soluble, sorbed, 71 acetate, microbial biomass, and microbial residues. Carbon, nitrogen and phosphorous may move among these states within each of four organic matter-microbe complexes: plant litterfall, animal 72 73 manure, particulate organic matter, and humus. Microbial biomass in ecosys is an active agent of 74 organic matter transformation. The rate at which each component is hydrolyzed is a function of substrate concentration that approaches a first-order function at low concentrations, and a zero-75 76 order function at high concentrations. These rates are regulated by soil temperature through an 77 Arrhenius function and by soil water content through its effect on microbial concentration. Similar to the growth and decline of vegetation biomass described above, the net change in 78 79 microbial biomass is determined by the difference between heterotrophic respiration and maintenance respiration. When heterotrophic respiration is greater than maintenance respiration, 80 81 the excessive amount of respiration is used as growth respiration that drives microbial growth 82 according to the energy requirements of biosynthesis.

84 Supplemental Material Table 1. Key soil properties of the (a) palsa (b) bog (c) fen at Stordalen

85 Mire used in *ecosys*.

|           | Depth | BD                 | K sat | TOC                | TON                | FC             | WP           | pН  |
|-----------|-------|--------------------|------------------|--------------------|--------------------|----------------|--------------|-----|
|           | m     | mg m -3 | $mm h^{-1}$      | g kg -1 | g kg -1 | $m^{3} m^{-3}$ | $m^3 m^{-3}$ |     |
| (a) Palsa |       |                    |                  |                    |                    |                |              |     |
|           | 0.01  | 0.10               | 100              | 452.04             | 8.88               | 0.4            | 0.15         | 3.9 |
|           | 0.05  | 0.10               | 100              | 438.38             | 9.62               | 0.4            | 0.15         | 3.9 |
|           | 0.1   | 0.12               | 25               | 388.16             | 10.90              | 0.4            | 0.15         | 3.9 |
|           | 0.2   | 0.20               | 25               | 343.97             | 12.21              | 0.4            | 0.15         | 3.9 |
|           | 0.3   | 0.30               | 25               | 331.83             | 13.86              | 0.4            | 0.15         | 4.1 |
|           | 0.4   | 0.80               | 20               | 304.80             | 14.19              | 0.4            | 0.15         | 4.5 |
|           | 0.5   | 1.20               | 18               | 208.73             | 10.89              | 0.4            | 0.15         | 4.4 |
|           | 0.6   | 1.20               | 15               | 206.77             | 10.88              | 0.4            | 0.15         | 4.4 |
|           | 0.7   | 1.23               | 13               | 203.92             | 10.77              | 0.4            | 0.15         | 5.1 |
|           | 0.9   | 1.25               | 12               | 200.71             | 11.10              | 0.4            | 0.15         | 5.3 |
|           | 1.1   | 1.25               | 12               | 150.00             | 8.60               | 0.2            | 0.11         | 5.3 |
|           | 1.3   | 1.35               | 10               | 120.00             | 7.60               | 0.2            | 0.11         | 5.3 |
|           | 1.5   | 1.35               | 10               | 120.00             | 7.60               | 0.2            | 0.11         | 5.3 |
| (b) Bog   |       |                    |                  |                    |                    |                |              |     |
|           | 0.01  | 0.02               | 500              | 390.20             | 4.22               | 0.4            | 0.15         | 4.2 |
|           | 0.05  | 0.02               | 500              | 407.15             | 5.59               | 0.4            | 0.15         | 4.2 |
|           | 0.1   | 0.04               | 500              | 403.20             | 6.81               | 0.4            | 0.15         | 4.2 |
|           | 0.2   | 0.04               | 500              | 418.90             | 8.83               | 0.4            | 0.15         | 4.2 |
|           | 0.3   | 0.15               | 300              | 461.90             | 11.93              | 0.4            | 0.15         | 4.2 |
|           | 0.4   | 0.35               | 200              | 466.60             | 13.06              | 0.4            | 0.15         | 4.4 |
|           | 0.5   | 1.05               | 100              | 466.20             | 13.30              | 0.4            | 0.15         | 4.6 |
|           | 0.6   | 1.25               | 60               | 406.20             | 13.30              | 0.4            | 0.15         | 4.7 |
|           | 0.7   | 1.30               | 50               | 406.20             | 13.30              | 0.4            | 0.15         | 4.8 |
|           | 0.9   | 1.33               | 40               | 406.20             | 13.30              | 0.4            | 0.15         | 4.9 |
|           | 1.1   | 1.35               | 25               | 400.00             | 13.60              | 0.2            | 0.11         | 5.0 |
|           | 1.3   | 1.35               | 15               | 400.00             | 13.60              | 0.2            | 0.11         | 5.0 |
|           | 1.5   | 1.35               | 15               | 400.00             | 13.60              | 0.2            | 0.11         | 5.0 |
| (c) Fen   |       |                    |                  |                    |                    |                |              |     |
|           | 0.01  | 0.02               | 500              | 436.90             | 13.47              | 0.4            | 0.15         | 5.7 |
|           | 0.05  | 0.02               | 500              | 435.18             | 14.97              | 0.4            | 0.15         | 5.7 |
|           | 0.1   | 0.04               | 500              | 435.14             | 15.55              | 0.4            | 0.15         | 5.7 |
|           | 0.2   | 0.04               | 500              | 380.83             | 15.55              | 0.4            | 0.15         | 5.7 |
|           | 0.3   | 0.15               | 300              | 340.83             | 14.47              | 0.4            | 0.15         | 5.7 |
|           | 0.4   | 0.35               | 200              | 336.51             | 16.49              | 0.4            | 0.15         | 5.7 |
|           | 0.5   | 0.70               | 100              | 336.51             | 17.65              | 0.4            | 0.15         | 5.7 |

| 0.6 | 1.10 | 60 | 430.21 | 22.65 | 0.4 | 0.15 | 5.7 |
|-----|------|----|--------|-------|-----|------|-----|
| 0.7 | 1.20 | 50 | 430.21 | 22.65 | 0.4 | 0.15 | 5.8 |
| 0.9 | 1.25 | 40 | 430.51 | 22.65 | 0.4 | 0.15 | 5.9 |
| 1.1 | 1.30 | 25 | 430.51 | 22.60 | 0.2 | 0.11 | 6.0 |
| 1.3 | 1.35 | 15 | 380.00 | 20.60 | 0.2 | 0.11 | 6.0 |
| 1.5 | 1.35 | 15 | 380.00 | 20.60 | 0.2 | 0.11 | 6.0 |

Abbreviations BD: bulk density, Ksat: saturated hydraulic conductivity, TOC: total organic carbon, TON: total organic nitrogen, FC: field capacity, WP: wilting point.

90 Supplemental Material Figure 1. Annual mean air temperature (a) and precipitation (b) extracted

91 from GSWP3 (blue), CRUNCEP (red), and ECMWF (yellow) at the Stordalen Mire.

92

---

## Author Response (AR2)

The authors would like to thank the editor for the valuable comments and suggestions to improve the clarity of our manuscript. We have carefully revised our manuscript based on the suggestions provided by the editor. The comments and suggestions have been taken into account in the revised manuscript, as follows (original editor's comments in bold):

**Specific comments:**

**Line 73-74: Jones et al. (2017) and Lundin et al. (2016) have contrary findings. Are peatlands sinks or sources in the long term? Can you add a sentence to clarify the ambiguity, and further justify the need for work such as the present study?**

The estimates reported in Lundin et al. (2016) were based on compiling carbon fluxes currently measured across a subarctic catchment, which does not provide information regarding carbon budget projection over peatlands. However, Lundin et al. (2016) highlighted the importance of spatial heterogeneity on high latitude carbon budget estimates. The sentence has been revised to improve its clarity (Line 71-74).

**Line 116: "thaw depth (in the active layer at palsa and bog sites or in the seasonally-frozen layer at fen sites)". ALD is the maximum thaw depth above permafrost. Anything less than that is a thaw depth, as is thaw in seasonally frozen ground. You model thaw depth progression. Please go through the manuscript and make sure you verify each instance of ALD: do you really mean ALD or thaw depth in each case? Your Figure 4 presents thaw depths, and I think that is correct.**

We thank the editor for bringing up the issue. We have checked and corrected our uses of ALD in the revised manuscript.

**Line 134-136: Please add a few sentences more about the physiography of the study region (e.g., topography, regional geology, when deglaciated, permafrost thickness, near-surface permafrost temperatures, permafrost temperature change rates in the region?). This will give more context and relevance to the study.**

Some of the requested information of the study region has been added accordingly to improve of study site description (Line 136-147).

**Line 136 New paragraph**

We have started a new paragraph here, as suggested by the editor (Line 148).

**Line 144 Change to "however". This will highlight the difference in the direction of change.**

We have changed "and" to "however", as suggested by the editor (Line 156).

**Line 146 New paragraph and Line 151 Append to previous paragraph.**
**Begin sentence with " At present, the Stordalen..."**

We have adjusted the format in this section to apply the editor's suggestions (Line 159-166).

**Line 154 Begin a new paragraph here.**

We have adjusted the format in this section to apply the editor's suggestion (Line 167-178).

**Line 186-188 change "period" to periods; "days 87-146" to "60 days (28 March (day 87) to 27 May (day 147))"; "and days 148-341" to "to 193 days (28 May (day 148) to 7 December (day 341))"; and rewording Line 188**

We have applied the edits suggested by the editor (Line 199-202).

**Line 232-233 "aboveground" to "above-ground"; insert "that are allowed to change with changing environmental conditions"; "belowground" to "below-ground"**

We have applied the edits suggested by the editor (Line 246-248).

**Line 236-237 "patch-scale" to "patch scale"; "landscape-scale" to "landscape scale"**

We have applied the edits suggested by the editor (Line 251-252).

**Line 239-240 Reviewer 2 asked for the vegetation parameters to be included in the manuscript. Please add a table to the main document or add it to the supplemental material and reference it.**

We have included the vegetation parameters in our Supplemental Material Table 2, which is referenced in Line 305-307.

**Line 342-347 "depth" to "depths"; "was" to "were"; "becomes greater than" to "exceeds"; "The" to "In contrast, the"; "by the end of July" to "nearly one month earlier"**

We have applied the edits suggested by the editor (Line 358-363).

**Line 351 You responded to Reviewer 2's question about consistnet over-predictions of net CO2 uptake for bog, but did not provide the explanation in the revised manuscript. This is needed in the discussion.**

The over-predictions of net CO2 uptake in the bog and the corresponding explanation have been included in the revised manuscript (Line 379-383).

**Line 372-374 insert "that is"; insert ". This may be"; insert ":"; insert ";"**

We have applied the edits suggested by the editor (Line 391-393).

**Line 382 Reviewer 2 asked for clarification regarding water table simulation and lateral water transport. You responded to the set of questions, but did not revise the manuscript to provide explanations to new readers. Please revise the manuscript or supplemental materials accordingly.**

We have included an explanation of the limitations of water table simulated in our one-dimensional column simulation in the revised manuscript (Line 411-415). We have added a description of our water table simulation in our *ecosys* qualitative description in the supplemental material (Line 39-42).

**Line 409 "weaker" to "comparatively lesser"**

We have applied the edits suggested by the editor (Line 430).

**Line 418 Reviewer 2 questioned this. I think that you should add some text to clarify the reasoning behind this choice, to head off further questions.**

We have included the reason behind the choice of our seasonality calculation (Line 439-441).

**Line 432 This is still a part of Results and Discussion. Please adjust numbering accordingly. EG. this will become 3.4, and the sub-section will become 3.4.1**

We have applied the edits suggested by the editor (Line 454).

**Line 433 Reviewer 1 wanted to know if there was a talik at any of the peatland sites. It is an important question because talik development is common in degrading peatlands, and microbial activity may continue in the unfrozen zoneF See my**

**comment on figure 1. You don't know if a talik developed or not, but you should bring up the issue. You should add in a sentence or two that treats your lack of knowledge about talik development in the field, the ability of the model to capture talik develoment, and the implications on your results and discussions.**

We have added a section discussing the limitations of our field measurements and the associated implications on our results in the revised manuscript (Line 456-464).

**Line 449 Do you really mean thaw-depth development?**

The ALD has been replaced with thaw depth in the revised manuscript (Line 480).

**Line 570 Section #4 now, not #5**

We have applied the edits suggested by the editor (Line 601).

**Line 938 Please add a note to clarify how the %ages are calculated (the rules); what the % coverage means. Can't tell without going to the main text.**

We have added a note to clarify the definition of the data coverage (Line 972-973).

**Line 940 Not part of the title. make this a note below the table.**

We have made the sentence "RRMSEs are relative root mean squared errors." as a note below the table (Line 976).

**Line 944-945 Not part of the title. Make this a note below the table.**

We have made the sentence "All gas exchanges are in units of g C m-2." as a note below the table (Line 980).

**Figure 1 Permafrost is typically considered impermeable. The water table has to remain above the permafrost table.**

We have corrected the water table position drawn in Figure 1.

**Line 967 Do you actually not know the active-layer depth at the bog site? If so, say explicitly on line 159. Based on the measured field data, it looks as if there is either no permafrost, or there is a talik with permafrost much lower than 90 cm. Again, this points out the necessity to discuss taliks and related implications.**

We do not know the ALD at the bog site because it is deeper than the maximum depth of

our thaw depth measurements. We have included a new section discussing taliks and related implications (Line 456-464).

**Line 968 "deepens below" to "exceeds"**

We have applied the edits suggested by the editor (Line 1002).

**Line 978 "depth" to "depths"**

We have applied the edits suggested by the editor (Line 1011).